# UserIdentifier: Implicit User Representations for Simple and Effective Personalized Sentiment Analysis

**Fatemehsadat Mireshghallah**[1]*, **Vaishnavi Shrivastava**[2], **Milad Shokouhi**[2],
**Taylor Berg-Kirkpatrick**[1], **Robert Sim**[3], **Dimitrios Dimitriadis**[3]
[1] University of California San Diego, [2] Microsoft Corporation, [3] Microsoft Research
[fatemeh, tberg]@ucsd.edu,
[vashri,milads,rsim,didimit]@microsoft.com

## Abstract

Globally federated models are trained to be as generalizable as possible, with user invariance considered desirable since the models are shared across multitudes of users. As such, these models are often unable to produce personalized responses for individual users, based on their data. Contrary to widely-used personalization techniques based on meta and few-shot learning, we propose UserIdentifier, a novel scheme for training a single shared model for all users. Our approach produces personalized responses by adding fixed, non-trainable user identifiers to the input data. We empirically demonstrate that this proposed method outperforms the prefix-tuning based state-of-the-art approach by up to 13%, on a suite of sentiment analysis datasets. We also show that, unlike prior work, this method needs neither any additional model parameters nor any extra rounds of few-shot fine-tuning.

## 1 Introduction

Federated learning is a form of distributed learning where data never leaves each user's device (Wang et al., 2021; Konečnỳ et al., 2018; Mireshghallah et al., 2020). Instead, the user trains a model on their device locally, and then shares the gradients (model updates) with a centralized server, which aggregates the gradients from different users and sends the updated model back to all of them, for further training. Personalization arises in applications where different clients need models specifically customized to their environment and profiles (Yang and Eisenstein, 2017; Mazaré et al., 2018; Flek, 2020). For example, a next-word- prediction task applied on the sentence " I live in ... ", requires prediction of a different answer, customized for each user (King and Cook, 2020). This need for customization in federated learning stems from the inherent heterogeneity existing in the data and the labels of different clients, especially when the task is classification (Kulkarni

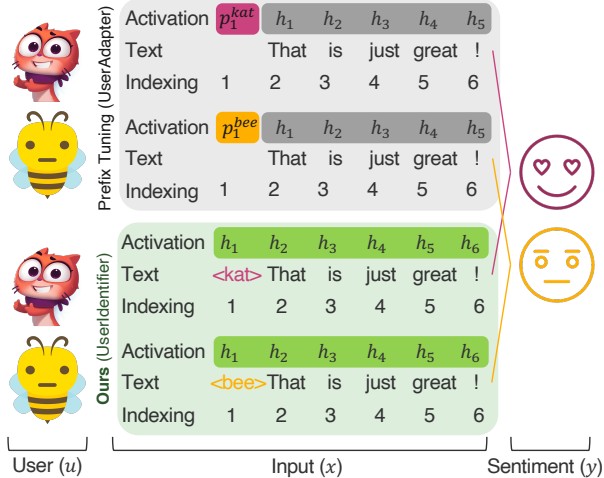

Figure 1: An overview of the proposed method, UserIdentifier, compared to its prefix-tuning counterpart. $p_1^u$ denotes the trainable prefix vector for user $u$, in the prefix tuning method. UserIdentifier, on the other hand, does not have trainable user-specific parameters and uses static text ("<kat>" and "<bee>") to condition a globally trained model (green), for each user.

et al., 2020; Wang et al., 2018). Figure 1 shows an example of the sentence "That is just great!". This sentence could carry a positive sentiment, a neutral apathetic sentiment, or even a completely negative sentiment. A non-personalized model cannot correctly predict the label for the different users.

Most techniques for personalization generally involve two phases: first, a global model is built between all users, and then, it is personalized for each client using their data (Kulkarni et al., 2020; Schneider and Vlachos, 2019; Lee et al., 2021). In such cases, each user has either an entirely separate model, or additional personal parameters, causing significant overheads, both in terms of storage of the large models, and the computation complexity of training separate models for each user. User-Adapter (Zhong et al., 2021), the state-of-the-art in sentiment analysis personalization, takes a prefix-tuning based approach (Li and Liang, 2021) to address this problem, as shown in Fig. 1. In the first

---

* Work done as part of an MSR internship. Corresponding author email: fatemeh@ucsd.edu

phase, a global model is trained in a user-agnostic way on the full dataset. In the second phase, each user $u$ is assigned their own prefix vector, $p_1^u$, which is trained separately for them, on their data. If there are $N$ users, there would be $N$ separate rounds of training, producing $N$ vectors. During this prefix-tuning phase, the main transformer model is frozen and shared between users.

To alleviate these training and storage costs and also improve overall performance, we propose training a single, shared personalized model, which can capture user-specific knowledge by conditioning on a unique, user-specific sequence of tokens from the classifier's vocabulary. We name this sequence "user identifier", and dub the underlying method of adding user identifiers to the input UserIdentifier. This is shown in Fig. 1, where we add the randomly generated, and non-trainable user identifiers "`anka Sau`" and "`Beh KY`" to each user's sample, and then train the transformer classifier model, on these augmented samples. The user identifiers just use the underlying model's vocabulary and embeddings, and do not add any tokens nor any user embeddings to the model. They are also static over time, and unique to each user, which means the user "bee" in Fig. 1 will have "`Beh KY`" pre-pended to all their samples, and no other user has this identifier. This is similar to the prompting of models like GPT-3 (Brown et al., 2020), however, here the prompt is fixed and used as data augmentation during training, and the model is not generative. As such, we only do training once, and have one set of shared parameters for all users. The approach is similar in essence to that of (Daumé III, 2009), which augments each individual feature with domain annotations.

We experiment with different types of strings for user identifiers, such as real usernames from the dataset, consecutive numbers, random digits, random non-alphanumeric tokens and random tokens (all types) and observe that, surprisingly, random identifiers, sampled from all possible tokens in the vocabulary perform best, providing $1.5\% - 13\%$ classification accuracy improvement on average, over the prefix-tuning based method UserAdapter (Zhong et al., 2021). We also study different lengths of identifiers. We report our results on three different sentiment analysis datasets (Sentiment 140, IMDB, and Yelp).

## 2 UserIdentifier

In this section, we first explain how UserIdentifier operates, then we go over the parameterization and

Table 1: Dataset specifications

| Dataset | # Users | # Samples | # Classes |
|---|---|---|---|
| IMDB | 1,012 | 137,710 | 10 |
| Yelp | 4,460 | 428,369 | 5 |
| Sent140 | 1,100 | 56,557 | 2 |
| Sent140 (skewed) | 473 | 23,155 | 2 |

learning procedure.

UserIdentifier is a data augmentation method which consists of adding a sequence of user-specific tokens (user identifier, $u_{id}$, drawn from the tokenizer's vocabulary) to each sample, $x$, to provide user-related cues to the model and help it learn individual behaviour and preferences, all in one shared model. Figure 1 shows how this augmentation works. Each utterance is prepended by the user identifier to create the augmented sample $[u_{id};x]$, and then used as input to the model, for the training stage. There is no restriction on what the make-up or the length of the user identifier sequence can be, as long as it is not longer than the maximum sequence length the model can input. However, in practice, since the sequence length is shared with the textual content of the user's input, it is better that the identifier sequence is not too long, so as to not lose the data. We study different types of identifiers and ablate them in Sections 3.3 and 4.3.

For parameterizations of the user identifiers, we use parameter tying, where the user identifiers use the same set of parameters for their embeddings as the rest of the user's input text. The entire transformer model is being trained to minimize the cross-entropy loss for the classification, with training input $x$ augmented as $[u_{id};x]$ with its user id.

## 3 Experimental Setup

### 3.1 Tasks, Datasets, and Models

We evaluate the proposed method on the task of sentiment analysis. Table 1 shows a summary of the datasets used in our experiments. We use the IMDB (Diao et al., 2014) and Yelp (Tang et al., 2015) datasets for comparison with the UserAdapter method (Zhong et al., 2021) and for the ablation studies. Each user's data is split into train, test, and validation sets, with 0.8, 0.1, 0.1 ratios. For comparison purposes, we are using a subset of the available users, i.e. those with fewer than 50 samples, as done by (Zhong et al., 2021) in support of few-shot learning, for reporting test accuracy. As such, we report test accuracy on a test set of 229 users for the IMDB task, and on a

set of $1,213$ users for the Yelp task. We use the RoBERTa-base model for this set of experiments.

In addition to IMDB and Yelp, we also report the performance of the proposed method on the Sentiment140 dataset (Go et al.; Caldas et al., 2018), which is a set of Tweets collected from Twitter and labeled positive or negative based on the emojis in each Tweet. For this dataset, unlike with IMDB and Yelp, we report test accuracies on all users. We use the methodology provided by (Li et al., 2019) to preprocess and partition this dataset. We create a second version of this dataset, and mark it as "skewed". For this skewed data, the users have been selected such that their sentiments are mostly skewed, i.e. we only include users with $80\%$ or more positive or negative Tweets. We do this to create a setup where data is more heterogeneously distributed. We use BERT-base-uncased for evaluations on the Sentiment140 dataset.

## 3.2 Baselines

**Conventional Training.** Before investigating the UserIdentifier performance, we establish the baseline performance. Our first baseline is conventional fine-tuning of the pre-trained transformer model on the full dataset, without any user-level personalization.

**UserAdapter.** The second baseline, which is the most closely related to our work, is User-Adapter (Zhong et al., 2021). In UserAdapter, a per-user embedding is learnt through few-shot learning. These personal vectors are prepended to the users' data to create personal responses. In other words, this work proposes prefix-tuning (Li and Liang, 2021) on a user-level. Unlike our method, UserAdapter consists of two phases, as discussed in Section 1: the first phase of general model fine-tuning, where all of the available data is used to fine-tune the pre-trained model for a given task, and the second phase where each user's data is used to train their own personal vector. This means UserAdapter, unlike our method, requires adding separate, per-user trainable parameters to the model, and storing the trained value of those parameters for each user.

**Trainable User Embeddings.** UserIdentifier uses the same set of parameters (BERT embeddings) for embedding both the sample content, and the user identifiers. In other words, the text and user embedding parameters are tied. To untie these parameters, we introduce a third baseline, with trainable user embeddings. In this setup, while the tokens used for the user identifier are still drawn from the pre-trained model's tokenizer vocabulary, we're creating and training a separate set of parameters for the user embedding, instead of using the pre-trained model's embedding.

## 3.3 Types of User Identifiers

We investigate five scenarios (types of sequences) for the user identifiers. The length of the user identifier sequences can vary in terms of number of tokens ($L$) for the last three of these scenarios.

**Default (Def.)**: This scenario uses the real user id (e.g., username) of that user, when provided by the dataset and if they are not private. We only have this option available for the Sentiment140 dataset.

**Consecutive Numbers (Num.)**: We assign each user a unique number, from $1$ to $N$, representing each user (up to $N$ users).

**Random sequence of digits (Rand. Dig.)**: In this scenario, $L$ independent and identically distributed (i.i.d) samples from the set of digits ($0$ to $9$) are drawn, creating a sequence of length $L$ for each user.

**Random sequence of tokens with non-alphanumeric characters (Rand. Non.)**: $L$ i.i.d samples are drawn from a subset of tokens (with size $400$) that contain non-alphanumeric characters, e.g., the token Ã"". The motivation for this scenario is that such user identifiers might be easier for the model to distinguish from the text (if we make sure the textual content in the sample has no overlapping tokens with the identifier).

**Random sequence of all tokens (Rand. All)**: This scenario draws $L$ i.i.d samples from the set of all available tokens in the tokenizer vocabulary.

## 4 Results

In this section, we first show the performance gain of UserIdentifier, over conventional training. Then, we benchmark the proposed UserIdentifier performance against the baselines (since the baseline is a centralized method, we also apply UserIdentifier in a centralized way for this particular experiment, to have a fair comparison). Then, we ablate different scenarios for the user identifiers with varying lengths. In our experiments we observed that the models would converge faster if we add the user identifier to both the beginning and then end of the samples, so that is what is reported here.

### 4.1 Summary of Results

Table 4 shows the performance gain of applying UserIdentifier, in a federated setup. UserIdentifier can be readily applied in federated learning, by assigning identifiers to each user and then asking

Table 2: Comparison of sentiment classification accuracy of UserIdentifier, with the baselines of Section 3.2. Num., Def. and Rand. refer to the different types of user identifiers introduced in Section 3.3.

| | Dataset | Conventional | UserAdapter | Trainable User Emb. | | | UserIdentifier | | |
|---|---|---|---|---|---|---|---|---|---|
| | | | | Num. | Def. | Rand. All | Num. | Def. | Rand. All |
| RoBERTa | IMDB | 45.1 | 46.2 | 45.5 | – | 48.9 | 50.1 | – | **52.5** |
| | Yelp | 68.3 | 70.2 | 68.3 | – | 70.6 | 69.5 | – | **71.3** |
| BERT | Sent140 | 84.7 | – | 84.7 | 86.3 | 86.5 | 84.9 | 87.1 | **87.1** |
| | Sent140 (Skewed) | 86.3 | – | 87.2 | 89.3 | 90.0 | 87.5 | 90.3 | **90.4** |

Table 3: Effect of the length (in terms of #tokens and type (Section 3.3) of user identifier sequence on classification accuracy.

| | Seq. Len. | Rand. Dig | Rand. Non. | Rand. All |
|---|---|---|---|---|
| IMDB | 5 | 48.8 | 51.3 | 52.2 |
| | 10 | 47.4 | 51.7 | 52.5 |
| | 20 | 47.1 | 50.2 | 51.1 |
| | 50 | 46.5 | 48.7 | 50.8 |
| | 200 | 33.3 | 32.8 | 40.1 |
| Yelp | 5 | 68.6 | 69.3 | 70.8 |
| | 10 | 68.7 | 69.6 | 71.3 |
| | 20 | 68.4 | 68.6 | 71.0 |
| | 50 | 67.8 | 69.0 | 70.6 |
| | 200 | 63.2 | 60.2 | 65.1 |

them to append it to all their samples. We have used the Rand. All type of user identifier for this experiment, since we observed in previous sections that it was the most effective. In general, the baseline performance and the performance gain the federated setup is slightly lower than centralized learning, which is due to the distributed nature of FL, and the fact that only average of multiple gradient updates are shared with the server for aggregation.

### 4.2 Comparison with Centralized Baselines

A comparison of UserIdentifier with the state-of-the-art UserAdapter method, and the other baselines is presented in Table 2. For the **Num.** (consecutive numbers) and **Def.** (default username) scenarios, as detailed in Section 4.3, the length of the user identifier sequences depends solely on the tokenization process. For the case of **Rand. All** (randomly sampled from all vocabulary tokens), however, it is shown that the sequence length of 10 tokens provides the best performance through the ablation study, therefore the results are reported for this length. Since the default usernames for IMDB and Yelp datasets are not provided, the corresponding results are not reported here.

It is shown that UserIdentifier with randomly generated identifiers outperforms all baselines, in all tasks. Our intuition is that UserIdentifier outperforms

UserAdapter because of the collaborative learning and personalization which is happening simultaneously, unlike the case of UserAdapter where personalization is performed separately for each user. The performance of trainable user embeddings appears inferior to that of UserIdentifier, which could be attributed to the parameter tying used in UserIdentifier. This parameter tying couples the learning problems for both domains (user identifier and text) and allows us to jointly learn from the full data, as in (He et al., 2019). For the Sentiment140 dataset, we can see that increasing the heterogeneity or skew in the dataset boosts the benefits brought about by UserIdentifier. This shows that the proposed method performs better in setups where personalization is actually needed (Deng et al., 2020).

### 4.3 Ablation Studies

Table 3 shows our ablation study into the length and the type of the user identifier sequence, for IMDB and Yelp datasets. The most evident trend is that performance significantly degrades in both datasets when the length of the user identifier sequence exceeds 20 tokens, holding for all identifier types. This is because the length of the input text itself is essentially decreased (the maximum sequence length for RoBERTa is 512, and the textual content of the sample is truncated to fit the user identifier in), when increasing the length of the identifier. This decreases the useful information which could be used to infer sentiment, and in turn it has an adverse effect on accuracy.

Another observation is that randomly sampling from the tokenizer's entire vocabulary outperforms sampling only from digits or from the non-alphanumeric tokens. This can be attributed to the different sizes of the sampling spaces for these three types, and the probability of overlap in user identifier from user to user. For the random digits (**Rand. Dig.**) the sample space size for each token position is 10, the number of possible digits. For the non-alphanumeric tokens, we have limited them

to $400$, and for the token type all (**Rand. All**), the possible sample space is $47{,}400$. This means that the probability of having token overlaps in user identifiers is much much smaller in the last scheme, than it is for the other two.

## 5 Conclusion

In this work, we present a novel approach for learning global models, producing personalized classification responses. This method which doesn't require model extensions or specialized training algorithms, consists of appending a fixed, non-trainable, unique identifier string to each sample during training and inference.

## Ethical Considerations

Our proposed model is intended to be used for addressing the problem of personalization, by learning one shared model for all users, and querying it using a personal identifier. One potential measure that needs to be taken for deployment of such technology is to setup proper authentication tools, so that each user can only query with their own identifier and prevent users from breaching privacy by querying other users' models. However, this could be a concern in other personalization setups too.

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

## A  Appendix

Table 4: Performance of UserIdentifier for sentiment classification in a federated learning setup.

| | Dataset | Conventional | User Identifier |
|---|---|---|---|
| RoBERTa | IMDB | 44.30 | 47.23 |
| | Yelp | 68.40 | 70.60 |
| BERT | Sent140 | 84.40 | 86.30 |
| | Sent140 (Skewed) | 86.50 | 90.00 |

### A.1  Performance on Unseen Users

To measure how robust the proposed method is to new users that have never been seen before, we run evaluation on new users, and report the results in Table 5. For this experiment we have used the best models from Tables 2, and tested them on samples from new users, without appending any user identifiers. It is noteworthy that there is some distribution shift between these unseen users and the seen users from Table 2, especially for Yelp, as we used samples that were not used in the original training/test/val setup (this test set contains 5000 samples for Yelp and 1357 samples for IMDB).

The UserIdentifier column refers to accuracy of those datapoints on models trained with user identifiers, and the conventional column shows the accuracy but on a conventionally trained model, which would be the baseline. We can see that both models behave similarly, which suggests that for unseen datapoints, the UserIdentifier trained model falls back to a conventional model, and does not behave even worse.

### A.2  Further User-level Accuracy Studies

Figure 2 shows the change in user accuracy, when we use UserIdentifier for training, instead of conventional training for each user. In other words, the horizontal axis shows $conventional_{acc} - UID_{acc}$ for each user, and the vertical axis shows the count of users.

As the plots show, on average across the two datasets, $32.1\%$ of the users see improvements in accuracy, whereas $54.2\%$ don't see any change.

### A.3  Maximally Distant User Identifiers

To better understand the effect of edit distance between user identifiers, We also experimented with **maximally distanced** identifiers (for the Rand. All setup), where the maximum distance would be the length of the identifier here, since each token in the identifier can take substantially large number of values. For this experiment, we used rejection

Table 5: Evaluation results on unseen users.

| | UserIdentifier Accuracy (%) | Conventional Model Accuracy (%) |
|---|---|---|
| IMDB | 50.4 | 50.9 |
| Yelp | 50.1 | 49.8 |

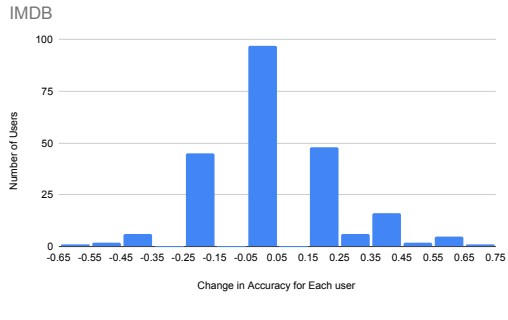

(a) IMDB

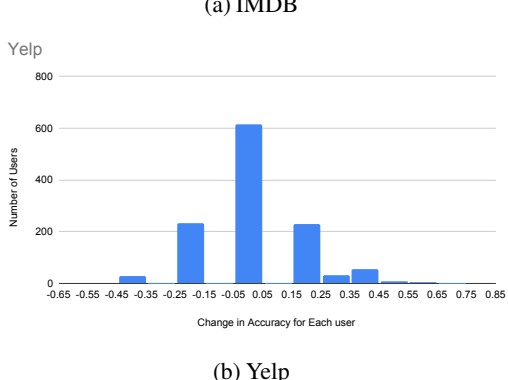

(b) Yelp

Figure 2: Distribution of test accuracy **change** across users.

sampling for user ids, as in if a new random sampled had any token overlaps with existing user ids, we would reject it and sample a new one. We observed results very similar to the ones with the random identifiers, which we hypothesize is because the random identifiers are already highly distanced and rarely overlap (less than $10\%$ of the users have non-maximal distance).