# OpenReview forum: "UserIdentifier: Implicit User Representations for Simple and Effective Personalized Sentiment Analysis "
_aclweb.org/ACL/2022/Workshop/FL4NLP — FL4NLP@ACL2022_

### Official Review · Reviewer_RU41 · 2022-03-23

**Rating:** 6
**Confidence:** 4

**Review:**

**Summary**

The paper proposes a data augmentation method to handle personalized prediction for text classification problems.

**Overall comments**

The method is simple and seems to work well on the toy problems studied in the paper. The experiments are adequate for a workshop paper (but can be improved to provide more insight into the performance). The writing is clear. The paper is on topic. I list drawbacks below.

**Cons**
- The paper compares different methods on simple datasets for the task of sentiment classification. To make the empirical study more compelling, one may consider additional tasks (language generation) and datasets (GLUE, table2text, dialog, summarization).
- It's unclear how the method improves upon baselines with better pre-trained models, e.g., Roberta*-large*.

**Other suggestions**
- The method could be combined with federated learning.
- The paper explores different ways to create the user-identifier in text format, but one could easily imagine the user-identifier being prompt embeddings directly -- randomly sample high-dimensional gaussians for each user as their prompt embedding.

---

### Official Review · Reviewer_wnZb · 2022-03-26

**Rating:** 7
**Confidence:** 3

**Review:**

## Summary

This paper proposes learning a personalized sentiment analysis model from the text by appending or prepending a user-specific string (termed "UserIdentifier") to the input text. Then a single transformer model is finetuned on data from all the individuals. Incorporating user identifiers help learn a better and more personalized model for each individual. The proposed method is compared with three other approaches --- finetuning with original data, finetuning with original data followed by prefix-tuning, and finetuning with trainable user identifiers. The authors justified their choice of selecting user-identifiers by appropriate ablation experiments. The "UserIdentifier" approach outperforms other baselines on Yelp, Sent140, and IMDB datasets.

## Strengths:
- Although the solution builds upon recent findings that demonstrate parameter efficient finetuning/few-shot learning by prompting with task-specific texts or introducing trainable input embeddings, the idea of introducing user-specific strings is interesting.
- I appreciate that the authors discussed different ways to assign user-identifiers and tried to partially explain the best choice in Sec A.3 and 4.3.
- The paper also studied generalization to new users briefly. Interestingly, the model performance is almost similar to finetuning with no user-identifiers (though slightly lower), thus providing personalization without hurting.


## Scope for Improvement:
- **Federated vs. Centralized Setups**:
Since we are eventually interested in a federated setup and personalized models, one of the baselines would be training a model per user, which is missing. While this may not be parameter efficient, each user will train its model on their local machine saving massive communication costs and maybe using similar or compute. The authors should discuss this scenario in the paper at least.
- **Writing**:
     - a. The main paper is mainly motivated by federated learning & need for personalized models, but the experiments are performed in a centralized setup which is ok. However, this is not clarified until sec 4. It would be nice to have this clarified in the introduction.
     - b. Sec 4.3 last paragraph ignores the L parameter in the discussion. The overlap will be much less even with just L=2 and sample space=400.
Minor: Table 4 is in the appendix. Either use a different numbering convention or add a small note in brackets that it is in the appendix.
- **Trainable Embeddings**:
It is counterintuitive that fixed prefixes outperform trainable embeddings, as Li & Liang (2021) and Hambardzumyan et al. (2021) show that it outperforms fixed prefixes. Even though the above references are not in the same context, ideally, more flexibility should help improve the model training. This raises the question if this can be explained by overfitting? Did the author compare training performance for these models?
The authors argue in the paper that simultaneous adaptation of parameters hurts learning. Would further "embedding only training" of the "UserIdentifier" approach improve or maintain performance?

### Refs:
- WARP: Word-level Adversarial ReProgramming (Hambardzumyan et al., ACL 2021)
- Prefix-Tuning: Optimizing Continuous Prompts for Generation (Li & Liang, ACL 2021)

---

### Official Review · Reviewer_XFKB · 2022-03-27
**Simple idea, strong empirical performance. Rigorous evaluation including interesting ablation analysis.**

**Rating:** 8
**Confidence:** 4

**Review:**

## Strengths
- Simple approach, trivial to implement
- Strong empirical performance
- Thorough empirical evaluation: multiple benchmarks (including skewed version of Sent140), ablation over token lengths and types of user-ids, performance of unseen users
- Privacy preserving. Rand.All can be implemented locally with no privacy loss (Def. and Num. cannot, but they don't work as well)

## Weaknesses
- No comparison against some other popular FL personalization schemes (like Ditto). UserAdapter is the only comparison, while something like Ditto or pFedMe are more established
- It is not clear why trainable user embeddings perform worse (it is very unintuitive, at least to me). Authors mention that "coupling learning problems in both domains is useful", but a deeper analysis might help.
- Their intuition on why UserIdentifier outperforms UserAdapter (collaborative learning and personalization is happening simultaneously). I don't understand how UserIdentifier performs any collaborative learning. Assume UserA and UserB behave "similarly", and a method that does collaborative learning might learn this and exploit it. In UserIdentifier, A and B would get random ids - so no collaborative learning would happen
- Large, over-parameterized models like RoBERTa-base or BERT-base are not practical in FL (on-device constraints). Would their scheme work in smaller models, where the embeddings are smaller and less over-parameterized?

## Suggestions
- Consider expanding to harder FL-NLP problems outside of sentiment classification (e.g, LM)
- Does this scale to a large #users, say millions? As the #users increase, almost all token embeddings will be modified by some user?
- An interesting setting would be running UserIdentifier in the same setting as UserAdapter: personalization as a separate task after global training
- On 'unseen' users: Another interesting setting would be to run local run train + eval on unseen users (instead of just eval, as you do)

---

### Decision · Program_Chairs · 2022-03-26

Accept